# Effect of Varying Dietary Crude Protein Level on Feed Intake, Nutrient Digestibility, Milk Production, and Nitrogen Use Efficiency by Lactating Holstein-Friesian Cows

**DOI:** 10.3390/ani10122439

**Published:** 2020-12-19

**Authors:** Constantine Bakyusa Katongole, Tianhai Yan

**Affiliations:** Sustainable Agri-Food Sciences Division, Agri-Food and Biosciences Institute, Large Park, Hillsborough, County Down, Northern Ireland BT26 6DR, UK; Tianhai.Yan@afbini.gov.uk

**Keywords:** dietary crude protein, milk yield, milk N efficiency, total mixed ration

## Abstract

**Simple Summary:**

Over the past decades, European Union countries have been under increasing pressure to reduce nitrogen pollution resulting from agricultural activities and ensure compliance with environmental legislations aimed at reducing the amount of nitrogen emitted into the atmosphere. Among the various forms of nitrogen losses into the atmosphere from agriculture, losses associated with farmed livestock (particularly dairy cows) are top priority. The nitrogen losses associated with livestock originate mostly from the excreta (feces and urine), with dietary protein content as a major driver. Despite the concern about dietary protein, the feeding of diets with excess protein relative to requirements (protein overfeeding) is evident on many commercial dairy farms in Europe, with the belief that it enhances milk production and in part for purposes of providing a margin of safety.

**Abstract:**

The effect of dietary crude protein (CP) level on intake, digestibility, milk production, and nitrogen (N) use efficiency was studied. Twenty-four Holstein-Friesian cows (17 multiparous and seven primiparous) were grouped by parity, days in milk, milk yield, and live weight into six blocks of four, and randomly assigned to four total mixed ration (TMR) treatments, containing 141, 151, 177, or 210 g CP/kg dry matter (DM), over 28 day experimental periods. Apparent total-tract DM and fiber digestibilities and milk fat composition were similar across treatments. Milk protein and urea-N compositions, and urinary and manure N excretion increased linearly, while milk N efficiency (MNE) decreased linearly with increasing CP. DM intake was highest with the 177 diet, while CP intake increased linearly with increasing CP, peaking at 200 g/kg DM. Milk yield increased with CP intake for the three lower CP levels, peaking at 176 g CP/kg DM. The further increase in CP level from 177 to 210 g/kg DM did not result in improved milk yield, but resulted in decreased milk N secretion and increased urinary N excretion. In summary, milk protein composition increased linearly with increasing CP, accompanied by a linear decrease in MNE, resulting in a bell-shaped relationship between milk yield and dietary CP level.

## 1. Introduction

Maximizing milk production while minimizing protein input has been one of the extremely important tasks in dairy cow nutrition in recent years. Firstly, protein is a considerably expensive nutrient; hence, feeding excess protein relative to requirements (i.e., protein overfeeding) results in unnecessary feeding expenses and, hence, low dairy farm profitability. Protein overfeeding can also result in low dairy farm profitability because of reduced milk nitrogen (N) efficiency [1]. As the dietary protein level increases beyond that needed to meet requirements, feed N use efficiency reduces, whereby milk N secretion reduces and the excretion of urinary N increases [2,3]. Secondly, dietary protein is one of the major drivers of environmental pollution with N from dairy farming [3,4,5]. Arriaga [6] reported that only 25.8% (range 19.2% to 32.3%) of the total feed N consumed by cows on commercial dairy farms is secreted in milk, with the majority of the remainder being excreted about equally in feces and urine. Powell [7] reported an average of 25.4% (range 18.2% to 32.6%). In spite of these environmental and economic concerns, protein overfeeding is rather a common occurrence under modern dairy production, particularly on farms feeding total mixed rations [6,8]. In addition to the inaccuracies associated with some protein models for balancing protein requirements of lactating dairy cows, the tendency on the part of dairy farmers to feed higher crude protein (CP) diets than recommended has been mentioned to potentially lead to protein overfeeding [8].

Over the past two decades, considerable research efforts have been directed at reducing N pollution from livestock production in response to increased environmental concerns and legislations regarding emissions from agriculture. Livestock excreta (urine and feces) can contribute to N pollution of the environment as ammonia and nitrous oxide (through volatilization, nitrification, and denitrification processes), nitrate leached to ground water, and N runoff to surface water [9,10,11]. Various studies have assessed the effect of dietary CP level on N use efficiency and reported a nearly twofold increase in urinary N excretion when dietary CP contents were increased from 15.1% to 18.4% [1], 16.2% to 20.1% [12], 14.8% to 16.7% [13], and 14.9% to 17.5% [14]. Thus, reducing dietary protein content should decrease excretion of the environmentally labile urinary N. However, the strategy of reducing dietary protein content is only viable if milk production remains economically profitable. Furthermore, substantial evidence has revealed that dietary N use efficiency (consequently, the extent of N excretion in urine and feces) by dairy cows at a given level of N intake is influenced by a number of factors, including animal variation (i.e., genetic variability for digestive or metabolic efficiency). Thus, the present study assessed N utilization and excretion using Northern Irish dairy cows. This study evaluated the effect of feeding total mixed ration (TMR) diets containing varying levels of CP on feed intake, nutrient digestibility, milk production, and N use efficiency by lactating Holstein-Friesian cows.

## 2. Materials and Methods

This study was conducted at the Agri-Food and Biosciences Institute, Hillsborough (Northern Ireland, UK). The study was approved by the Ethical Review Committee of the Institute and was in accordance with the Animals (Scientific Procedures) Act 1986.

### 2.1. Experimental Animals

Twenty-four lactating Holstein-Friesian cows (17 multiparous and seven primiparous) were selected from the autumn-calving dairy herd at Agri-Food and Biosciences Institute, Hillsborough, UK. The cows were between 146 and 200 days in milk (DIM), averaging 2.4 parity (standard deviation (SD) 1.5), 645 kg live weight (SD 57), and 32 kg of milk/day (SD 6) at the time of enrolment in the study.

### 2.2. Treatment Diets and Experimental Design

The study involved four total mixed ration (TMR) treatment diets that were prepared from perennial ryegrass (*Lolium perenne* L.) silage and two concentrate meals (i.e., low protein and high protein). The two concentrate meals were supplied premixed containing 123 and 220 g CP/kg (fresh basis) for the low- and high-protein concentrate meals, respectively. The chemical compositions of the ryegrass silage and the two concentrate meals that were used to make the TMR treatment diets are presented in Table 1. The silage and concentrate were mixed in a ratio of about 48:52 (dry matter (DM) basis) targeting dietary CP levels of 141, 151, 177, or 210 g/kg DM. The chemical compositions of the TMR treatment diets are presented in Table 2.

The 24 cows were grouped into six blocks of four cows on the basis of parity, days in milk, pretreatment milk yield, live weight, and body condition score. The cows within each block were randomly allocated to one of the four TMR treatment diets. Thereafter, the 24 cows were randomly paired up (i.e., 12 cow-pairs) ensuring that no cow-pair consisted of cows allocated to the same treatment diet (Figure 1). Two cows (i.e., one cow-pair) were enrolled in the study at a time (in the order from pair 1 to pair 12, at 3 day intervals) for 28 day experimental periods. The first 20 days were for adaptation to the treatment diets, followed by 5 days of individual feeding (for collecting nutrient balance data), and finally 3 days in two respiration chambers for measuring gas emissions. It is important to point out that, in addition to nutrient balance assessment, this study had an additional objective of measuring gas emissions using two respiration chambers. It was for this reason that the study was designed to enroll the experimental cows in pairs (results of the gas emissions objective are not included here).

### 2.3. Feeding Management and Measurements

During the 20 day adaptation period, the cows were group-fed their TMR treatment diets. The decision to group-feed the cows was because of the lack of personnel (required for the daily management of individual animals) and there were not enough individual tie-stalls to house that number of cows. The respective TMR diets were mixed separately (for each treatment) using a feeder wagon (Vari-Cut 12, Redrock, Armagh, Northern Ireland), and tipped into a series of feeders mounted on weigh scales. Individual cow access to the feeders was programmed (cows were able to only access feeders containing their rightful treatment diets) by means of an electronic “neck-tag” identification system (Controlling and Recording Feed Intake (CRFI), BioControl, Rakkestad, Norway). This made it possible to monitor the individual cow intakes on a daily basis. During the 5 day individual feeding period, the cows were housed in individual tie-stalls with continuous access to fresh water. The head of each cow was loosely tied at a halter in the individual tie-stall. The respective TMR diets were mixed separately (for each cow) using a power concrete mixer. The cows were fed once daily at 9:00 a.m., and then any refusals (orts) and spilled feed were collected and weighed on the following day at about 8:00 a.m. (before new feed was offered). The amount of TMR offered was adjusted daily on the basis of the previous day’s intake to ensure 10% orts.

Day 1 of the 5 day individual feeding periods was for adaptation, and the final 4 days were for data collection (milk yield, feed intake, and total feces and urine). The daily TMR intakes were manually recorded. The cows were milked twice daily (a.m. and p.m.) using the milking apparatus installed in the tie-stalls, and the individual milk yields were recorded manually. The milk yields were adjusted to 4% fat-corrected milk (4% FCM). Daily a.m. and p.m. milk samples were collected and preserved with an antimicrobial preservative (Broad Spectrum Microtabs^TM^ II: Advanced Instruments Inc., USA) in a refrigerator (at 5 °C). At the end of each 5 day individual feeding period, the a.m. and p.m. milk samples were bulked, and a composite sample was analyzed for protein, fat, lactose, casein, and urea-N compositions using an infrared milk analyzer MilkoScan Combifoss™7; Foss Electric, Hillerød, Denmark).

For the 24 h total collections of feces and urine, a Velcro patch (vinyl fabric, about 15 cm × 15 cm) was designed with a hole to correspond to the vulva opening. The patch was glued (EVO-STIK 528 Instant Contact Adhesive: Bostik Ltd., Stafford, UK) to each cow around the perineal region (area under the tail) to facilitate collection of urine separately from feces. The Velcro patches were supported by two straps (one on either side) that were glued to the rump. For each patch, Velcro tape (loop-side) was sewn around the hole corresponding to the vulva, which helped to hold in position a urine collection unit (with hook-side Velcro tape) connected to an 80 mm polyvinyl chloride (PVC) Layflat Hose (for channeling the urine into a plastic collection container). The feces of each cow were collected in a large plastic collection tray that was placed behind each tie-stall. To prevent ammonia from volatilizing, urine was collected after adding approximately 300 mL of 50% sulfuric acid to each urine collection container daily. The total daily amounts of feces and urine produced by each cow (24 h total) were weighed daily at 8:00 a.m. and recorded. About 2% of the day’s feces and 500 mL of the day’s urine were sampled daily and stored in a refrigerator (at 5 °C) for subsequent analysis. At the end of each 5 day individual feeding period, the daily fresh fecal samples of each cow were bulked (on an equal weight basis) and thoroughly mixed into a composite sample of about 1 kg. For each cow, a subsample of about 100 g was analyzed for total N content using wet chemistry analysis. The rest of the fresh sample was oven-dried at 60 °C for 144 h for DM content determination. The fecal DM content was then used to calculate the total daily fecal DM output for each cow. The oven-dried fecal samples were ground through a 1 mm sieve and analyzed for neutral detergent fiber (NDF) and acid detergent fiber (ADF) [16], as well as ash concentrations.

Two samples (about 200 g each) of the ryegrass silage and one sample (about 200 g) of the concentrate meals used were taken daily. One of the two silage samples and the concentrate meal samples were oven-dried at 60 °C for 48 h for DM content determination. The DM content (silage and concentrate meals) was used to calculate the total daily DM intake of each cow (from the daily TMR intakes). The other silage sample was stored fresh in a refrigerator (at 5 °C). At the end of each 5 day individual feeding period, the daily oven-dried samples were ground through a 1 mm sieve, bulked (on an equal weight basis), and the composite samples were analyzed for total N, NDF, ADF, and ash content. Similarly, the daily fresh silage samples were bulked (on an equal weight basis), and composite samples were analyzed for total N, pH, ammonia-N, and volatile fatty acid concentration using wet chemistry analysis. The DM content and analyzed chemical compositions of the silage and the two concentrate meals were used to calculate the chemical compositions of the TMR diets (Table 2).

### 2.4. Statistical Analysis

All data were analyzed using SAS (version 9.1, SAS Institute, Cary, NC, USA, 2003). Treatment differences were considered significant when *p* ≤ 0.05 and tendencies when 0.05 < *p* ≤ 0.10. The effect of dietary CP level on feed intake, nutrient digestibility, milk production, and nitrogen use efficiency was analyzed using the PROC MIXED procedure for repeated measures. The repeated measure was day, and the variable that uniquely defined the subjects was cow within order of enrolment in the study (cow-pair). A first-order autoregressive covariance structure was used, while the residual maximum likelihood (REML) was used as the estimation method. The statistical model included block, diet, day, and diet × day interaction as fixed effects. No covariate was included in the statistical model for the feed intake, milk components (fat, lactose, protein, casein, and urea-N) yield, and excreta N data. Pretreatment milk yield was included as a covariate for the milk yield and milk N secretion data, while pretreatment milk yield, DIM, and parity were included as covariates for the apparent total-tract digestibility data. Polynomial contrasts were used to detect linear and quadratic effects of dietary CP level. Selection of the fixed and covariate effects depended on whether their inclusion in the model resulted in a smaller Akaike information criterion (AIC) and/or made a meaningful improvement in treatment means and standard errors.

The effect of dietary CP level on milk components was analyzed using the PROC MIXED procedure. Block and diet were included in the statistical model as fixed effects. DIM was included as a covariate. The effect of cow within the order of enrolment in the study (cow-pair) was considered a random factor. Linear and quadratic effects of dietary CP level were also estimated.

## 3. Results and Discussion

### 3.1. Feed Intake and Apparent Total-Tract Digestibility

Intakes of DM, organic matter (OM), NDF, and ADF were affected by dietary CP level in a quadratic (*p* < 0.05) manner, being higher (*p* < 0.05) with the 177 diet, but not different between the 141, 151, and 201 diets (Table 3). Quadratic functions predicted peak intakes of DM, OM, NDF, and ADF to occur at dietary CP levels of 176, 175, 173, and 173 g/kg DM, respectively (an average dietary CP level of 174 g/kg DM). Intake of CP was affected by dietary CP level in a linear (*p* < 0.05) and quadratic (*p* < 0.05) manner. The intake of CP first increased linearly with increasing dietary CP level to a peak value and then decreased. A quadratic function predicted the peak value to occur at a dietary CP level of 200 g/kg DM. As was the case in the current study, Broderick [2] also reported a linear increase in DM intake as dietary CP was increased from 15.1% to 16.7% and then to 18.4%. However, in contrast, Mutsvangwa [15] reported no dietary CP level effects on the intakes of DM, OM, NDF, and ADF when lactating cows were fed TMR diets containing either 14.9% or 17.5% CP. Olmos Colmenero and Broderick [17] also reported no dietary CP level effect on DM intake when CP levels increased from 13.5% to 15.0%, 16.5%, 17.9%, and then to 19.4% CP.

Apparent total-tract DM, OM, NDF, and ADF digestibilities did not differ (*p* > 0.05) across dietary CP levels and averaged 714, 734, 606, and 612 g/kg, respectively (Table 3). This result appears to contradict some previous studies with TMR-fed dairy cows. Olmos Colmenero and Broderick [17] reported quadratic increases in apparent DM, NDF, and ADF digestibilities as dietary CP was increased from 13.5% to 15.0%, 16.5%, 17.9%, and then to 19.4% CP, while Groff and Wu [18] reported linear and quadratic increases in ADF digestibility as dietary CP was increased from 15.0% to 16.25%, 17.5%, and then to 18.75% CP. However, although Broderick [2] reported linear increases in apparent digestibilities of NDF and ADF as dietary CP was increased from 15.1% to 16.7% and then to 18.4%, this author observed no dietary CP level effect on apparent DM digestibility. Unsurprisingly, the apparent total-tract digestibility of CP increased linearly (*p* < 0.05) with increasing dietary CP level, being highest with the 201 diet, intermediate with the 177 diet, and lowest with the 141 or 151 diets. Increased dietary protein level results in increased dilution of the metabolic fecal N, thus yielding a greater apparent digestibility of CP [19,20]. Similarly, Broderick [2] also reported a linear increase in N digestibility as dietary CP content was increased from 15.1% to 18.4%, while Olmos Colmenero and Broderick [17] reported linear and quadratic increases in CP digestibility as dietary CP content was increased from 13.5% to 19.4%.

### 3.2. Milk Yield, Components and Milk Components Yield

Milk yield was affected by dietary CP level in a quadratic (*p* < 0.05) manner, with no further increase from 177 to 201 g CP/kg DM (Table 4). A quadratic function predicted peak milk yield to occur at a dietary CP level of 176 g/kg DM. There was a tendency for higher (*p* = 0.094) 4% FCM yield with the 177 diet, showing a trend (*p* = 0.079) toward a quadratic response to increasing dietary CP level. A quadratic function predicted peak 4% FCM yield to occur at a dietary CP level of 177 g/kg DM. Feed efficiency, expressed as either milk yield or 4% FCM per kg of feed DM consumed, was not affected by dietary CP level (*p* > 0.05) and averaged 1.28 and 1.40, respectively. The lack of dietary CP level effect on feed efficiency suggested that the higher milk yield observed with the 177 diet was at least partially attributable to its higher DM intake.

In the present study, increasing the dietary CP level beyond 177 g CP/kg DM did not result in improved milk yield. This result is in agreement with studies by Burgos [21] and Borucki Castro [13], who reported decreases in milk yield when dietary CP increased from 17% to 21% and from 19.7% to 20.1%, respectively. This lack of benefits for milk yield observed in the present study when the dietary CP was increased beyond 177 g/kg DM could be explained by the less efficient ability to salvage and recycle urea-N to the rumen when ruminants are fed excessively high-protein diets. Urea-N that is recycled to the rumen acts as a precursor-N for microbial protein synthesis for subsequent digestion and absorption across the small intestine [22,23]. Several studies [15,24,25] have indicated that total urea-N salvaged and recycled to the rumen decreases as dietary protein level increases. Characteristically, high dietary CP levels result in increased ruminal ammonia-N concentration [23,24], which has been reported to be inversely related to urea-N transfer to the rumen [15,26]. Thus, increasing the dietary CP level beyond 177 g/kg DM obviously resulted in increased ruminal ammonia-N concentration, which ended up being excreted in urine rather than being salvaged and reutilized.

Milk fat and lactose compositions did not differ (*p* > 0.05) across dietary CP levels and averaged 46.1 and 47.9 g/kg, respectively (Table 4). Similarly, other previous studies [2,15] also reported no dietary CP level effect on milk fat composition. However, some other studies reported contrasting results. Olmos Colmenero and Broderick [17] reported a linear increase in milk fat composition when TMR diets ranging from 13.5% to 19.4% CP were fed to cows. In another study, Law [27] reported a linear decrease when TMR diets ranging from 114 to 173 g CP/kg DM were fed. It should be noted, however, that the milk fat composition observed in the present study was inexplicably high. Our mean milk fat composition (46.1 g/kg) was higher by 17.1% than the mean of 38.2 g/kg reported by Law [27] for cows (151 to 305 DIM) fed TMR diets with a comparable concentrate-to-forage ratio as the one fed in the present study. The lack of dietary CP level effect on milk lactose composition is consistent with other studies [2,15,27], where increasing dietary CP levels did not affect milk lactose composition of cows fed TMR diets with concentrate-to-forage ratios comparable to the one used in the current study.

Milk protein and casein compositions increased linearly (*p* < 0.05) with increasing dietary CP level, being highest with the 177 or 201 diets (Table 4). Similarly, Broderick [2] also reported a significant dietary CP level effect on milk protein composition as dietary CP was increased from 15.1% to 16.7% and then to 18.4%. However, in contrast, some other studies [15,17,27] observed no dietary CP level effect on milk protein composition. Milk urea-N composition was affected by dietary CP level in a linear (*p* < 0.05) and quadratic (*p* < 0.05) manner, being highest with the 201 DM diet, intermediate with the 177 diet, and lowest with the 141 or 151 diets. The observed linear increase in milk urea-N composition was anticipated because milk urea-N composition is related positively to dietary protein content or intake [18,21,28]. Thus, milk urea-N composition has been proposed as a diagnostic of protein feeding in dairy cows [6,28]. Since ruminal ammonia-N increases with increasing dietary protein level [23,24], an increase in milk urea-N concentration has to be expected when dietary protein level increases. Other studies [14,15,17] also reported greater milk urea-N compositions from cows fed high-protein diets compared with those fed low-protein diets. The milk urea-N composition observed in the current study with the 201 diet (205 mg/kg) reflected protein over-feeding. Melendez [29] classified milk urea-N values beyond 17 mg/dL as high. A milk urea-N value of approximately 11.7 mg/dL was reported to be indicative of satisfied N requirements of rumen microbes for grass silage-based diets [28].

Milk yields of fat and lactose did not differ (*p* > 0.05) across dietary CP levels, and averaged 1.17 and 1.22 kg/day, respectively (Table 4). In contrast, Broderick [2] and Law [27] reported linear and/or quadratic increases in milk yield of fat as dietary CP level was increased. There was a tendency for higher milk yields of protein and casein (*p* = 0.061 and *p* = 0.051, respectively) with the 177 or 201 diets showing linear (*p* < 0.05) responses to increasing dietary CP level. Milk yield of casein also showed a trend (*p* = 0.070) toward a quadratic response to increasing dietary CP level. A quadratic function predicted peak milk yield of casein to occur at a dietary CP level of 182 g/kg DM. This result is in agreement with studies by Broderick [2] and Law [27] who reported linear and/or quadratic increases in milk yield of protein as dietary CP level was increased. Milk yield of urea-N was affected by dietary CP level in a linear (*p* < 0.05) and quadratic (*p* < 0.05) manner. Milk yield of urea-N was affected by dietary CP level in a linear (*p* < 0.05) and quadratic (*p* < 0.05) manner, being highest with the 177 or 201 diets. A quadratic function predicted the peak milk yield of urea-N to occur at a dietary CP level of 187 g/kg DM. Similarly, Broderick [2] also reported a linear increase in milk yield of urea-N as dietary CP was increased from 15.1% to 18.4%.

### 3.3. Milk N Secretion, and N Excretion in Feces and Urine

Urine output was affected by dietary CP level in a linear (*p* < 0.05) manner and showed a trend (*p* = 0.054) toward a quadratic response to increasing dietary CP level, being highest with the 177 or 201 diets (Table 5). A quadratic function predicted peak urine output at a dietary CP level of 187 g/kg DM. Urinary N excretion increased linearly (*p* < 0.05) with increasing dietary CP level, being highest with the 201 diet, intermediate with the 177, and lowest with the 141 or 151 diets. The proportion of N intake appearing as urinary N increased linearly (*p* < 0.05) with increasing dietary CP level, being highest with the 177 or 201 diets. The observed linear increases in urine output and urinary N excretion were anticipated because, when dietary RDP is in excess of the amount required by ruminal microorganisms, the protein is degraded to ammonia-N, absorbed, metabolized to urea in the liver, and lost in the urine [30]. However, urinary urea excretion requires water [19,31], which inevitably leads to higher water intake and, hence, increased urine output. When dietary protein (and the subsequent N intake) is not in excess, ruminal ammonia-N production is reduced, which leads to reduced urinary N. Others studies also reported linear increases in urine output and urinary N excretion with increasing dietary CP level [2,17,18] or higher urine output and urinary N excretion with higher CP diets than lower CP diets [15,32].Fecal DM output tended to be higher (*p* = 0.051) with the 177 g CP/kg DM and averaged 5.73 kg/day (Table 5). However, fecal N excretion was affected by dietary CP level in a quadratic (*p* < 0.05) manner (Table 5), being highest with the 177 diet, but not different among the 141, 151, and 201 diets. A quadratic function predicted peak fecal N excretion at a dietary CP level of 176 g/kg DM. This result is in agreement with studies by Broderick [2], Groff and Wu [18], Olmos Colmenero and Broderick [17], and Recktenwald [23], who reported linear and/or quadratic increases in fecal N excretion as dietary CP level was increased. However, in contrast, some other studies [3,14,32] reported no significant effect of dietary CP level on fecal N excretion. The proportion of N intake appearing as fecal N decreased linearly (*p* < 0.05) with increasing dietary CP level, being highest with the 141 or 151 diets, intermediate with the 177 diet, and lowest with the 201 diet.

Total manure N (fecal and urinary N) excretion increased linearly (*p* < 0.05) with increasing dietary CP level, being highest with the 177 or 201 diets (Table 5). This result was anticipated because the principal driver of N excretion is N intake [33]. In addition, several researchers [34,35] reported positive correlations between manure N excretion and dietary protein supply or N intake. The proportion of N intake appearing as manure N did not differ (*p* > 0.05) across dietary CP levels and averaged 75.2%. This was not surprising because increasing dietary CP level shifts the N excretion pattern from feces to urine. In the present study, as dietary CP level was increased from 141 to 151, 177, and then to 201 g/kg DM the proportion of N intake appearing as fecal N reduced by 8.3, 13.0, and 28.4 percentage units, respectively, while the proportion of N intake appearing as urinary N increased by 3.9, 14.8, and 35.3 percentage units, respectively. The average values observed in the present study for the proportion of N intake appearing as manure N are within the range reported in the literature for commercial dairy farms using TMR diets [6,7].

Milk N secretion was affected by dietary CP level in a linear (*p* < 0.05) and quadratic (*p* < 0.05) manner (Table 5). Milk N secretion first increased linearly with increasing dietary CP level to a peak value, with no further increase from 177 to 201 g CP/kg DM. A quadratic function predicted the peak value to occur at a dietary CP level of 183 g/kg DM. The proportion of N intake appearing as milk N (milk N efficiency) decreased linearly (*p* < 0.05) with increasing dietary CP level, being lowest with the 201 diet, and highest with the 141 or 151 diets (Figure 2). Compared with the lowest CP level (141 g CP/kg DM), the two highest CP levels (177 and 201 g CP/kg DM) resulted in 13.9 and 21.3 percentage unit reductions, respectively, in milk N efficiency. Similarly, other studies [2,3,15,32] also reported significantly lower milk N efficiencies with higher-CP diets than lower-CP diets. The observed linear decrease in milk N efficiency with increasing dietary CP level was anticipated because, when RPD is in excess of the amount required by ruminal microorganisms (as was the case with the two higher-CP diets), an increased ruminal ammonia-N concentration arises [23,24], which is known to inhibit urea-N transfer to the rumen [15,26], leading to increased urinary N excretion and, hence, reduced milk N efficiency. The milk N efficiency values observed in the current study were consistent with the data of other studies [2,15,17,32] with cows fed diets ranging from 14.9% to 19.4% CP.

### 3.4. Optimal Dietary CP Level

Quadratic regression functions and predicted optimal dietary CP levels for variables that showed quadratic responses (*p* ≤ 0.10) to increasing dietary CP level are summarized in Table 6. The optimal dietary CP levels were predicted by taking first derivatives of the quadratic equations. The optimal dietary CP levels for milk yield and milk N secretion averaged 180 g CP/kg DM.

## 4. Conclusions

In this study, varying dietary CP level from 141 to 151, 177, and then to 201 g/kg DM did not affect apparent total-tract DM, NDF, and ADF digestibilities, as well as milk fat composition and yield, by lactating Holstein-Friesian cows in mid lactation (146 to 200 DIM). However, apparent total-tract CP digestibility, milk protein and urea-N compositions, urinary N excretion, and manure N excretion increased linearly, while milk N efficiency decreased linearly with increasing dietary CP level. Intake of DM, milk yield, and fecal N excretion were affected by dietary CP level in a quadratic manner, with no further increase from 177 to 201 g CP/kg DM. Milk N secretion first increased linearly with increasing dietary CP level, peaking at a dietary level of 183 g CP/kg DM, and then decreased. It was clearly evident that increasing the dietary CP level beyond 177 g/kg DM did not result in benefits for intake of DM and milk yield, but resulted in increased urinary N excretion (an environmentally labile form of N), as well as decreased milk N secretion and, hence, reduced milk N efficiency. Milk N secretion was reduced by 7.5%, while milk N efficiency was reduced by 8.5 percentage units as dietary CP was increased from 177 to 201 g/kg DM.

## Figures and Tables

**Figure 1 animals-10-02439-f001:**
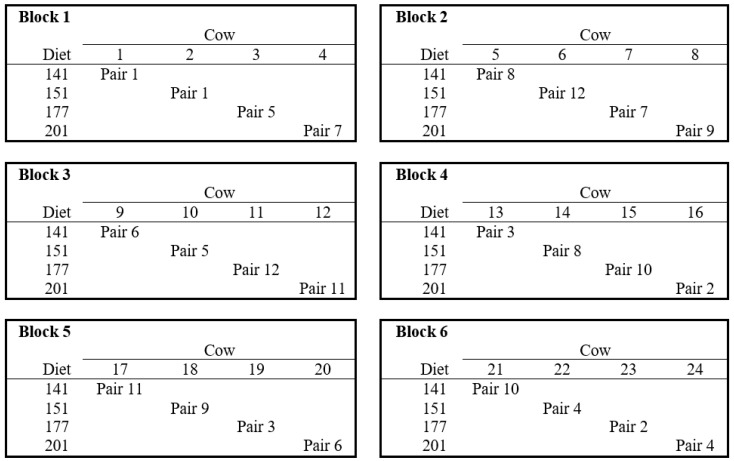
Layout for the six blocks of four cows each (including how they were paired up) and four treatment diets. Pair numbers also refer to the order in which the cows were enrolled in the study. Two cows (i.e., one pair) were enrolled at a time in the order from pair 1 to pair 12, at 3 day intervals.

**Figure 2 animals-10-02439-f002:**
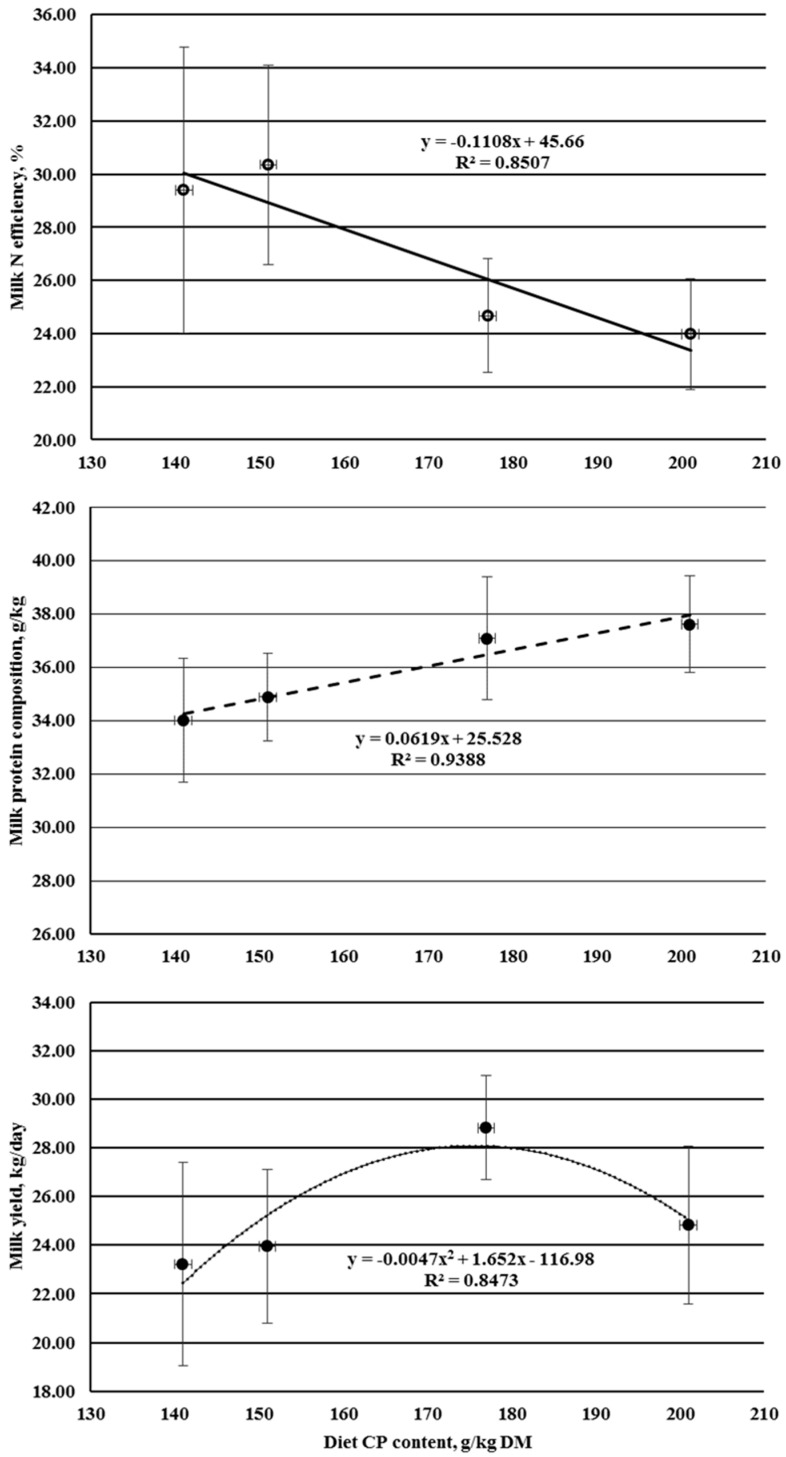
Increasing dietary CP level with milk yield, protein composition, and N efficiency. Bars represent the standard deviation of the means.

**Table 1 animals-10-02439-t001:** Chemical composition of the silage and concentrates used. DM, dry matter; CP, crude protein; NDF, neutral detergent fiber; ADF, acid detergent fiber.

		Concentrate	
	Ryegrass Silage	Low Protein	High Protein
Ingredient, % DM			
Wheat		24.5	22.0
Barley		23.5	21.9
Sugar beet pulp		20.2	2.0
Maize gluten		12.5	4.0
Soya bean hulls		8.1	7.6
Calcium salts of palm fatty acid		2.7	1.6
Rapeseed meal		2.0	10.2
Soya bean meal feed dehulled		2.0	26.3
Calcium carbonate		1.4	1.4
Acid Buf ^1^		1.4	1.4
Sodium chloride		0.7	0.7
Super dairy ^2^		0.5	0.5
Magnesium oxide		0.3	0.3
ActiSaf ^3^		0.1	0.1
Chemical composition			
DM, g/kg	336 ± 19.3	906 ± 5.2	903 ± 2.8
CP, g/kg DM	154 ± 8.9	134 ± 2.9	252 ± 1.5
Rumen-degraded protein (RDP) ^4^, g/kg CP	765	688	693
Rumen-undegraded protein (RUP) ^4^, g/kg CP	235	312	307
NDF, g/kg DM	554 ± 16.7	283 ± 10.5	243 ± 5.7
ADF, g/kg DM	314 ± 12.9	134 ± 3.6	106 ± 10.6
Water-soluble carbohydrates (WSC), g/kg DM	104 ± 1.1	55 ± 5.9	79 ± 1.9
Ash, g/kg DM	81 ± 4.6	75 ± 4.2	81 ± 3.9
pH	4.6	-	-
Total ammonia N, % total N	6.6	-	-
Lactic acid, %DM	12.2	-	-
Acetic acid, %DM	3.8	-	-
Butyric acid, %DM	0.57	-	-

^1^ Calcified seaweed included as a rumen acidity buffer (AB Vista, Belfast, UK); ^2^ mineral and vitamin premix pack for dairy cows (Devenish Nutrition, Belfast, UK), containing: vitamin A (9000 IU/kg), vitamin D3 (2000 IU/kg), vitamin E as alpha-tocopherol (75 mg/kg), biotin (20 mg/kg), iodine 7.5 mg/kg (11.90 mg/kg calcium iodate), cobalt 0.16 mg/kg (3.2 mg/kg granulated cobalt carbonate), copper 40 mg/kg (120 mg/kg cupric sulfate pentahydrate, 100 mg/kg cupric chelate of amino acid hydrate), manganese 50 mg/kg (64 mg/kg manganous oxide), zinc 100 mg/kg (104.175 mg/kg zinc oxide, 166.67 mg/kg zinc chelate of protein hydrolysates), and selenium 0.7 mg/kg (50 mg/kg selenium yeast); ^3^ live yeast Sc47 (*Saccharomyces cerevisiae*) for rumen health (Phileo, Belfast, UK); ^4^ feed into milk (FiM) system according to Thomas [15].

**Table 2 animals-10-02439-t002:** Ingredients and chemical composition of the total mixed ration (TMR) diets.

	Dietary CP Content, g/kg DM			
	141	151	177	201
Ingredient, % of TMR DM				
Ryegrass silage	47.3	47.9	47.7	47.5
Low protein concentrate	52.7	43.0	22.0	3.0
High protein concentrate	0.0	9.1	30.3	49.5
Chemical composition ^1^				
DM, g/kg	507	513	510	508
CP, g/kg DM	141	151	177	201
RDP, g/kg CP	103	109	127	144
RUP, g/kg CP	39	41	49	56
NDF, g/kg DM	411	408	399	388
ADF, g/kg DM	218	218	214	209
WSC, g/kg DM	81	82	86	90
Ash, g/kg DM	78	78	80	81

^1^ Calculated as formulated from the analyzed chemical compositions of the individual ingredients.

**Table 3 animals-10-02439-t003:** Effect of dietary CP level on intake and apparent total-tract digestibility.

	Dietary CP Content, g/kg DM				SEM	*p*-Value		
	141	151	177	201		Diet	Linear	Quadratic
DM intake, kg/day								
Silage	9.1 ^b^	9.1 ^b^	10.7 ^a^	9.4 ^b^	0.359	0.014	0.138	0.011
Concentrate	10.1 ^b^	9.9 ^b^	11.8 ^a^	10.4 ^b^	0.383	0.012	0.147	0.017
Total	19.2 ^b^	18.9 ^b^	22.5 ^a^	19.8 ^b^	0.730	0.011	0.136	0.013
OM intake, kg/day	17.7 ^b^	17.4 ^b^	20.7 ^a^	18.2 ^b^	0.665	0.012	0.152	0.013
CP intake, kg/day	2.72 ^b^	2.86 ^b^	3.98 ^a^	3.95 ^a^	0.132	<0.0001	<0.0001	0.015
NDF intake, kg/day	7.91 ^b^	7.72 ^b^	8.98 ^a^	7.68 ^b^	0.309	0.026	0.700	0.015
ADF intake, kg/day	4.19 ^b^	4.12 ^b^	4.83 ^a^	4.15 ^b^	0.183	0.039	0.516	0.021
Apparent total-tract digestibility								
DM, g/kg	707	719	715	715	13.1	0.890	0.773	0732
OM, g/kg	728	737	737	734	12.6	0.922	0.778	0.624
CP, g/kg	571 ^c^	611 ^b,c^	641 ^b^	695 ^a^	16.8	0.0002	<0.0001	0.928
NDF, g/kg	604	623	607	590	20.9	0.640	0.391	0.524
ADF, g/kg	613	626	618	591	20.6	0.556	0.289	0.414

SEM, standard error of least square means; OM, organic matter; ^a,b,c^ least square means in the same row with different superscript letters differ (*p* < 0.05).

**Table 4 animals-10-02439-t004:** Effect of dietary CP level on milk yield, components, and milk components yield.

	Dietary CP Content, g/kg DM				SEM	*p*-Value		
	141	151	177	201		Diet	Linear	Quadratic
Milk yield								
Milk, kg/day	23.7 ^b^	24.5 ^b^	28.6 ^a^	25.3 ^a,b^	1.29	0.048	0.153	0.044
4% FCM, kg/day	25.6	26.6	31.1	28.1	1.56	0.094	0.114	0.079
Milk/DIM	1.27	1.29	1.28	1.28	0.072	0.997	0.969	0.868
FCM/DIM	1.37	1.40	1.40	1.41	0.082	0.987	0.754	0.920
Milk components								
Fat, g/kg	45.6	45.9	45.5	47.5	0.779	0.254	0.121	0.252
Lactose, g/kg	47.8	47.9	47.7	48.3	0.600	0.910	0.682	0.627
Protein, g/kg	34.1 ^c^	34.8 ^b,c^	37.2 ^a,b^	37.5 ^a^	0.834	0.023	0.004	0.359
Casein, g/kg	27.1 ^b^	28.1 ^a,b^	28.7 ^a^	29.3 ^a^	0.475	0.029	0.005	0.445
Urea-N, mg/kg	105 ^c^	110 ^c^	189 ^b^	205 ^a^	4.04	<0.0001	<0.0001	0.003
Milk components yield								
Fat yield, kg/day	1.09	1.16	1.22	1.21	0.059	0.383	0.151	0.323
Lactose yield, kg/day	1.15	1.23	1.28	1.23	0.060	0.493	0.367	0.212
Protein yield, kg/day	0.81	0.89	0.99	0.96	0.049	0.061	0.022	0.116
Casein yield, kg/day	0.64	0.72	0.77	0.75	0.032	0.051	0.028	0.070
Urea-N yield, g/day	2.52 ^b^	2.99 ^b^	5.05 ^a^	5.22 ^a^	0.233	<0.0001	<0.0001	0.007

SEM = standard error of least square means; ^a,b,c^ least square means in the same row with different superscript letters differ (*p* < 0.05).

**Table 5 animals-10-02439-t005:** Effect of dietary CP level on milk N secretion, and N excretion in feces and urine.

	Dietary CP Content, g/kg DM					*p*-Value		
	141	151	177	201	SEM	Diet	Linear	Quadratic
Total N intake, g/day	436 ^b^	457 ^b^	637 ^a^	633 ^a^	21.1	<0.0001	<0.0001	0.015
Urinary excretion								
Urine output, kg/day	18.5 ^b^	18.6 ^b^	24.7 ^a^	24.1 ^a^	0.978	<0.0001	<0.0001	0.054
Urinary N, g/day	146 ^c^	156 ^c^	243 ^b^	287 ^a^	13.8	<0.0001	<0.0001	0.652
Urinary N, % N intake	33.7 ^b^	35.0 ^b^	38.7 ^a,b^	45.6 ^a^	2.58	0.008	0.001	0.501
Fecal excretion								
Fecal DM output, kg/day	5.61	5.23	6.39	5.7	0.275	0.051	0.174	0.119
Fecal N, g/day	185 ^b^	175 ^b^	234 ^a^	193 ^b^	11.8	0.016	0.110	0.023
Fecal N, % N intake	42.2 ^a^	38.7 ^a,b^	36.7 ^b^	30.2 ^c^	1.55	<0.0001	<0.0001	0.551
Manure N, g/day	331 ^b^	332 ^b^	480 ^a^	481 ^a^	22.7	0.0002	<0.0001	0.116
Manure N, % N intake	75.7	73.7	75.5	75.8	3.19	0.948	0.806	0.836
Milk N secretion								
Milk N ^1^, g/day	127 ^b^	138 ^a,b^	161 ^a^	149 ^a,b^	7.68	0.033	0.022	0.048
Milk N, % N intake	29.6 ^a^	29.9 ^a^	25.5 ^a,b^	23.3 ^b^	1.62	0.021	0.003	0.953

^1^ Determined as milk protein yield/6.38; SEM, standard error of least square means; ^a,b,c^ least square means in the same row with different superscript letters differ (*p* < 0.05).

**Table 6 animals-10-02439-t006:** Quadratic regression equations of dietary CP level (g/kg DM).

	Equation						Maximum ^1^	*p*-Value ^2^
Intake								
DM, kg/day	=	−55.380	+	0.876 CP	−	0.00249CP ^2^	176	0.013
OM, kg/day	=	−50.385	+	0.800CP	−	0.00228CP ^2^	175	0.013
CP, kg/day	=	−12.008	+	0.160CP	−	0.00040CP ^2^	200	0.015
NDF, kg/day	=	−18.823	+	0.318CP	−	0.00092CP ^2^	173	0.015
ADF, kg/day	=	−8.400	+	0.149CP	−	0.00043CP ^2^	173	0.021
Milk yield								
Milk, kg/day	=	−116.980	+	1.652CP	−	0.00470CP ^2^	176	0.044
4% FCM, kg/day	=	−133.510	+	1.860CP	−	0.00526CP ^2^	177	0.079
Milk components								
Urea-N, mg/kg	=	−940.340	+	11.152	−	0.02711CP ^2^	206	0.003
Milk components yield								
Casein yield, kg/day	=	−4.358	+	0.058CP	−	0.00016CP ^2^	182	0.070
Urea-N yield, g/day	=	−50.378	+	0.596CP	−	0.00159CP ^2^	187	0.007
N excretion								
Urine output, kg/day	=	−97.997	+	1.315CP	−	0.00352CP ^2^	187	0.054
Fecal N, g/day	=	−1290.370	+	17.312CP	−	0.04927CP ^2^	176	0.023
Milk N, g/day	=	−475.590	+	6.927CP	−	0.01893CP ^2^	183	0.048

^1^ Determined by taking the first derivative of the quadratic equation; ^2^ probability value of the quadratic effect of dietary CP level.

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
