# Peer review of "Effect of Varying Dietary Crude Protein Level on Feed Intake, Nutrient Digestibility, Milk Production, and Nitrogen Use Efficiency by Lactating Holstein-Friesian Cows"

_animals, 2020, doi:10.3390/ani10122439_

Round 1

Reviewer 1 Report

This is a well-written article on an important topic, namely, the impact of rising dietary CP levels on milk yield.

There are a few suggestions for improvements below.

Abstract:

You write

“ However, CP  intake, milk protein and urea N compositions, milk yields of protein and urea N, urinary and manure N excretion, and milk N increased linearly, while milk N efficiency decreased linearly.”

Linearly versus what parameter? CP Level? 

 “CP  intake, milk yield, milk urea N composition and yield, faecal N excretion, and milk N secretion increased quadratically.”

Again, versus what?

Those not entirely familiar with the lingo established in the nutritional sciences will ask: how can CP intake rise both linearly and quatratically with CP level?

It appears less confusing and more precise to write something like “CP intake first rose linearly and then fell with CP level in a roughly quadratic manner, reaching a peak at a level of …g/kg DM.”

Further on, it might also be better to simply write what you saw, e.g “In the first three feeding groups, milk yield rose with CP intake, reaching a maximum at 177 g of CP/kg of DM. The further increase in CP from 177 to 210 g of CP/kg of DM did not result  in improved milk yield, but resulted in decreased milk N and increased urinary N excretion.”

The last sentence might then simply read:

“ It was concluded that milk protein composition increased linearly in response to increasing dietary CP accompanied by a linear decrease in milk N efficiency, resulting in a bell-shaped dependency between milk yield and dietary CP content.“

Materials and Methods

Table 1: Please give details concerning “super dairy” and “ActiSaf”. Both are probably feed additives, but what is the producer? What is the composition?

Please explain the abbreviations “GE”. (Gross Energy?) OM (Organic matter?) and DM (Dry matter?); NDF, ADF at first mention

I assume that NH3-N refers to both NH3 and NH4+ N. I would suggest using Total Ammonia-N, since the word ammonia is generally used to designate both forms (NH3 and NH4+). NH3-N leaves one wondering if the NH3 content was analysed.  Or simply write “(NH3 & NH4+)-N”

Line 69: what does SD mean? (Standard deviation?)

Lines 72 ff: This is very confusing. It is not clear to me how you can achieve four different CP levels if silage and concentrate were always mixed in a ratio of 48:52.  Why don’t you add Table 2 at this point?

Line 102: Please give producer of Broad Spectrum MicroTabs  (containing a combination of Bronopol and Natamycin) and some explanaition of what it is.  Is it an antibiotic? (eg. “and preserved with an antimicrobial agent containing     Bronol and Natamycin (Microtabs, Bentley Instruments)”)

Line 103, 112, 124 etc in a publication, please use “refrigerator” and not the colloquial “fridge”.

Line 105: What company supplied the MilkoScan? (Yes, it can be googled but it is customary to give the origin in the M&M)

Line 107: how was the Velcro patch glued to the rump? Or perhaps you can provide a quote if the procedure has been established elsewhere?

Results and discussion

Line 160: Again, it is confusing when you write that Intake of CP increase both linearly and quadratically in response to increasing dietary CP level. 

Why not simply write what you observed, e.g. something like “rose with CP content, reaching a maximum at 176, after which levels declined. Statistically, data could be fitted with both a linear function (p < ….) and a quadratic function (p < ..). 

Line 257: Suggestion: “Urine output rose with dietary CP level,   being highest with the 177 or 201 g of CP/kg of DM diets. Data could be fitted both with a linear and a quadratic function (p < 0.05).

Line 293: Suggestion: “The dependency between milk N secretion and dietary CP level could be fitted both with a linear and a quadratic regression model, being lowest with the 141 g of CP/kg of DM diet”

Line 321: Suggestion: “Up to 177 g/kg of DM, increasing dietary CP level resulted in increasing  milk yield, milk urea N composition and yield, milk N secretion and faecal N excretion. “

Finally, I think that a graph showing the major findings of the study might help readers to grasp the central findings of the study, e.g. CP on x-Axis, milk protein composition, milk N efficiency, and milk yield on three Y-Axes.

Reviewer 2 Report

The research concerns a very important and current problems - rational animal nutrition and the impact of animal production on the environment. Intensive livestock production contributes to significant nitrogen emissions to the atmosphere, and the main factor influencing nitrogen emissions is the protein content in the diet. Therefore, one should strive to balance the diet in such a way as to minimize the negative impact on the environment.

Publication requires corrections. The scheme of assigning cows to individual experimental groups is incomprehensible. Please re-edit this fragment (79-89), preferably in the form of a table or diagram.

Please specify feeding in the adaptation period. Was the TMR the same for all groups or were there 4 different types of TMR with different protein content? It is not clearly specified, the reader must guess it.

Table 2 - error in the table header - the same CP level (g / kg of DM)

Please adapt 35 literature position to the requirements of the journal.

Reviewer 3 Report

Ruminant sustainable production and mitigation of negative charging impact of this branch of animal production on the natural environment currently is in the area of the interest of European Union policy and many institutions all over the world.  The authors of  the paper “Effect of Varying Dietary Crude Protein Level on Feed Intake, Nutrient Digestibility, Milk Production and Nitrogen Use Efficiency by Lactating Holstein-Friesian Cows” carried out and presented the results of the study that fit to this very important trend. The article was written carefully and clearly in English according to valid techniques for such kinds of paper. The feeding experiment design is simple but correct and gives the clear answers to study hypothesis. The material and methods of the trial have been presented in comprehensive and clearly way.

However, in my opinion the nutritional values of cow diets should be supplemented with data on WSC, RUP and RDP in relation to low protein and high protein concentrate and TMR-s that could affect the N management in cows feeding. Also I suggest to remove the information on GE value of diets. It does not introduce any relevant data to the nutritional value and is useless in diet formulation for ruminants.

All suggestions do not detract from the cognitive value of the work. It is a valuable bridgehead to knowledge in this area with applicable character. For this reason, that after making the small corrections by the Authors, I recommend for publication in Animals.

Reviewer 4 Report

The manuscript revealed the effect of varying dietary crude protein level on production of dairy cows. Nitrogen pollution is an important problem for the dairy cows’ feeding, and the level of dietary CP has closely relationship to the nitrogen emission. Therefore , it is meaningful to study the effect of dietary CP level on the nitrogen metabolism to find out the suitable dietary CP level. The manuscript found that although milk protein composition increased linearly in response to increasing dietary CP, milk yield increased quadratically accompanied by a linear decrease in milk N efficiency. The experimental design is logical, and the methods and results are clear. Some comments are as follows.

Table 1, the ingredient need to be calculated based on dry matter to make it comparable to other studies.

For statistical analysis, it is not clear how to deal with the 6 blocks. The effect of blocks need to be in the model as random or fixed effect.

Line 298-302, it is good to see the comparison of the data to other publication. But the author can give some explanations why the increase of dietary CP level enhances or reduces the N efficiency in different ranges of dietary CP levels.

Reviewer 5 Report

This study evaluated dietary crude protein level on feed Intake,nutrient digestibility, milk production and nitrogen efficiency by lactating dairy cows. In general, the manuscript was well written and data were well presented. However, this study lacks novelty because several similar studies can be found in literature. The authors needs to improve the introduction to emphasize the novelty of the study. In addition, one major flaw of experimental design is the group feeding of animals by dietary treatments in the first 20 days of experiment. Did animals compete with each other? What has been done to minimize sorting of feed ingredients? A justification of group feeding is needed.

SED in tables should be SEM, standard error of the mean.

Table 6 is almost not discussed.

The conclusion about recommended dietary protein level is too general without specifying animal lactation stage, milk production level, etc. The conclusion needs to more based on current experiment settings to be scientific sound.

Round 2

Reviewer 5 Report

The authors have addressed my comments adequately.